# Dethroning of Neuroendocrine Tumor as an Orphan Disease: US Incidence, Prevalence, and Survival in the 21st Century

**DOI:** 10.3390/cancers17203323

**Published:** 2025-10-15

**Authors:** Qian Yu, Fan Cao, Peter Hosein, Bin Huang, Paulo S. Pinheiro, Yating Wang, Jaydira Del Rivero, Gilberto Lopes, Aman Chauhan

**Affiliations:** 1Vascular and Interventional Radiology, Department of Radiology, University of Chicago Medical Center, University of Chicago, Chicago, IL 60637, USA; qian.yu@uchicagomedicine.org; 2School of Medicine, Georgetown University, Washington, DC 20007, USA; fc654@georgetown.edu; 3Division of Hematology/Oncology, University of Miami Miller School of Medicine, Miami, FL 33136, USA; phosein@med.miami.edu; 4Division of Cancer Biostatistics, Department of Internal Medicine, University of Kentucky College of Medicine, Lexington, KY 40536, USA; bhuan0@uky.edu; 5Division of Epidemiology & Population Health Sciences, Department of Public Health Sciences, University of Miami School of Medicine, Miami, FL 33136, USA; 6Hematology and Oncology, Ascension All Saints Hospital Cancer Center, Racine, WI 53405, USA; 7Center for Cancer Research, National Cancer Institute, Bethesda, MD 20814, USA; jaydira.delrivero@nih.gov; 8Sylvester Comprehensive Cancer Center, University of Miami Health System, Miami, FL 33136, USA; glopes@med.miami.edu; 9Division of Medical Oncology, Department of Medicine, University of Miami Miller School of Medicine, Miami, FL 33136, USA

**Keywords:** epidemiology, incidence, neuroendocrine tumor, survival

## Abstract

Neuroendocrine tumors (NETs) have historically been considered a heterogeneous but rare malignancy. This epidemiology study used national cancer registry data from 2000 to 2021 to examine trends in NET incidence, prevalence, and survival in the United States. The results showed that the number of new NET diagnoses has nearly doubled over this period, and more individuals are living with the disease. Survival has improved, particularly among patients with early-stage or low-grade tumors. These changes appear to be associated with improvements in diagnostic methods and increased disease recognition. The findings suggest that NETs are no longer rare and that the growing patient population may require expanded access to specialized care and research efforts.

## 1. Introduction

Neuroendocrine tumors (NET) encompass a rare and diverse group of tumors derived from neuroendocrine cells across the body [1]. While most NETs exhibit indolent behavior, those of higher grade and advanced stage often carry a poor prognosis [2]. Due to their rarity and variability, managing aggressive NETs typically necessitates referral to specialized NET centers employing a multidisciplinary approach. Collaboration across institutions is crucial for generating robust evidence through clinical trials, as single-institution studies are often underpowered. Therefore, population-based investigations like those utilizing the Surveillance, Epidemiology, and End Results (SEER) database are essential for understanding NET epidemiology and prognosis over time. The previous landmark study using SEER 18 registries (SEER-18) data up to 2012 covered less than 30% of the U.S. population [2]. Since then, the SEER program expanded to SEER 22 registries (SEER-22), encompassing nearly 50% of the population, with the most recently submitted data covering up to the year of 2021. Meanwhile, advancements in diagnostic methods such as gallium-68 DOTA-Tyr3-octreotate (68Ga DOTATATE) and copper-64 DOTA-Tyr3-octreotate (64Cu DOTATAE) have significantly improved diagnostics capabilities [3]. Together with the widespread use of endoscopy, computed tomography (CT), and magnetic resonance imaging (MRI), have further facilitated earlier detection of NETs, contributing to increased incidence, prevalence, and overall survival (OS) [3,4]. This study aims to provide contemporary insights into NET epidemiology in the 21st century U.S. by assessing its trends in incidence, prevalence, and OS using the updated SEER-22 database.

## 2. Materials and Methods

### 2.1. Data Collection

Data was retrospectively retrieved from the Surveillance, Epidemiology, and End Results Program 22 registry (SEER-22) and 17 registry (SEER-17) using SEER*Stat software, version 8.4.0.1, provided by the National Cancer Institute. Data submitted in November 2023 was used in the present study, which was accessed and analyzed in June 2024. SEER-22 and SEER-17 cover approximately 47.9% and 26.5% of the total U.S. population, respectively. Data analysis was conducted from June 2024 to July 2024.

### 2.2. Variable Definition

We designed our study based on previously established methodology by two landmark NET SEER analyses led by James Yao in 2008 and Arvind Dasari in 2017. Minor changes accounting for the NET WHO classification over the past two decades were incorporated. NET was identified using Site and Morphology codes based on the International Classification of Diseases for Oncology, 3rd Edition, according to prior literature [1]:

Site and Morphology.ICD-O-3 Hist/behav = ‘8150/0: Pancreatic endocrine tumor, benign (ICD-O-3 update)’, ‘8150/3: Pancreatic endocrine tumor, malignant’, ‘8151/3: Insulinoma, malignant’, ‘8152/3: Glucagonoma, malignant’, ‘8153/3: Gastrinoma, malignant’, ‘8154/3: Mixed pancreatic endocrine and exocrine tumor, malignant’, ‘8155/3: Vipoma, malignant’, ‘8156/3: Somatostatinoma, malignant’, ‘8240/2: Carcinoid tumor, in situ’, ‘8240/3: Carcinoid tumor, NOS’, ‘8241/3: Enterochromaffin cell carcinoid’, ‘8242/3: Enterochromaffin-like cell tumor, malignant’, ‘8244/3: Mixed adenoneuroendocrine carcinoma (ICD-O-3 update)’, ‘8245/3: Adenocarcinoid tumor’, ‘8246/2: Neuroendocrine carcinoma in situ’, ‘8246/3: Neuroendocrine carcinoma, NOS’, ‘8249/3: Atypical carcinoid tumor’.

Of note, “goblet cell carcinoid” and “goblet cell carcinoid in situ” were not used because this entity had been changed to “goblet cell adenocarcinoma”.

Age-adjusted incidence rate per 100,000 persons was calculated according to the 2000 US standard population, with annual percent change (APC) calculated with Joinpoint regression. Limited-duration prevalence per 100,000 persons was calculated for each year, defined as the sum of the yearly incidence and all the patients diagnosed in prior years; the 21-year duration prevalence was calculated in 2021. OS was measured using observed survival and the Kaplan Meier method. The median OS, 1-year OS, 3-year OS, and 5-year OS were calculated.

### 2.3. Statistical Analysis

SEER-22 was implemented to calculate incidence (including annual percentage change), limited-duration prevalence, and age-adjusted survival from 2000 to 2021, analyzed with SEER*Stat software according to methods described in a prior study [2]. Subgroup analysis was performed based on age, grading, disease stage, and primary site.

SEER-17 was used to perform multivariable analysis to identify factors associated with OS, as patient-level survival data could not be retrieved from SEER-22 to perform Cox proportional hazard ratio (HR) regression analysis. Statistical analysis was performed using StataMP 18.0 (STATA Corp., College Station, TX, USA). *p* values < 0.05 were considered statistically significant.

## 3. Results

### 3.1. Incidence

SEER-22 contains a total of 231,659 patients diagnosed from 2000 to 2021. The annual age-adjusted incidence of NETs increased from 4.6 per 100,000 in 2000 to 8.2 per 100,000 in 2021 (Figure 1, annual percentage change: 2.90 per 100,000, *p* < 0.05). There was a temporary decrease in incidence in 2020, approximately 7.3 per 100,000. Based on age, the incidence was the highest among patients older than 65 years, increasing from 18.0 per 100,000 in 2000 to 27.4 per 100,000 in 2021, as compared to the 1.2–3.0 per 100,000 increase among patients less than 50 years old and the 9.8–16.8 per 100,000 increase among patients 50–65 years old (Table 1).

With the exception of colorectal and liver NETs, the incidence rates across all primary sites trended up from 2000 to 2021 (Figure 2). The top 4 NET primary sites were consistent across the time frame, with colorectal at 1.46, small bowel at 1.45, and pulmonary at 1.43 per 100,000 in 2021. As the 4th highest incidence, pancreatic NET demonstrated a 5-fold increase from 0.29 in 2000 to 1.28 in 2021. Appendiceal NET also had a steep increase during 2010–2015, superseding gastric NET in 2015 as the 5th highest incidence in 2021, which was 0.97 per 100,000, as compared to its 0.078 in 2000.

Based on staging, the incidence of local disease has been the highest and increased the most, from 2.1 per 100,000 in 2004 to 4.3 per 100,000 in 2021, whereas it increased from 0.98 to 1.4 and 1.2 to 1.5 per 100,000 in regional and distant diseases, respectively (Figure 3). Based on grading, the incidence of G1 NET demonstrated the steepest and upward-trending curve, from 0.25 in 2000 to 4.7 per 100,000 in 2021. The incidence of G2 NET also increased from 0.14 to 0.87 per 100,000, whereas it remained relatively stable for G3/G4 between 2000 and 2016 in the range of 0.54 and 0.83, which slightly trended down afterwards to 0.41 in 2021.

### 3.2. Prevalence

From 2001 to 2021, the cumulative limited-duration prevalence increased from 0.0037% to 0.064% (Figure 4, Table 2). Among grade groups, prevalence increased the most in grade 1 NETs from 0.00023% in 2001 to 0.02993% in 2021, despite a temporary decrease during 2020. Based on primary sites, colorectal tumors were the highest among all locations (0.01675% in 2021). The prevalence of appendiceal tumor (0.00652% in 2021) crossed over and superseded stomach (0.00474% in 2021) in 2018 as the fifth highest prevalence after colorectal, small intestine (0.0127% in 2021), lung (0.01168% in 2021), and pancreas (0.00768% in 2021) (Figure 5).

### 3.3. Overall Survival

According to SEER-22 data, 1-year, 3-year, and 5-year OS for all patients diagnosed with NET were 77.4%, 66.1%, and 59.3%, respectively, while the median OS time was not reached (Figure 6, Table 3). Localized NETs demonstrated the most promising 5-year OS of 82.3%, compared to the 66.9% of regional disease and 26.1% of distant disease. According to grades (Figure 6, Table 3), OS was the worst among G3/G4 NETs with a 5-yr OS of 16.6%, whereas it was 80.0% and 66.5% for G1 and G2 NETs (Figure 7, Table 4). Appendiceal, small intestine, and colorectal NETs demonstrated the best 5-yr OS, which were 75.9%, 74.4%, and 72.1%, respectively (Table 3). Hepatic NET has the worst 5-yr OS of 28.2%, followed by pulmonary NETs (5-yr OS: 49.1%). According to the stage, the 5-yr OS ranged from 38.4% in liver NET to 88.2% in colorectal NET among local diseases. In regional diseases, the 5-yr OS ranged from 24.6% in liver NET to 78.1% in appendiceal NET. In distant diseases, the 5-yr OS ranged from 13.0% in pulmonary NET to 61.8% in small bowel NET.

The SEER-17 database includes 120,827 NET patients; patients diagnosed at a later time demonstrated improved survival (Table 5, all time periods, HR 0.92–0.97, *p* < 0.05, compared to 2000–2004). Patients diagnosed in 2015–2021 demonstrated an 8% reduced risk of death (HR, 0.92, 95% CI: 0.89–0.95, *p* < 0.001). Compared to patients with G1 NETs, G2 (HR, 1.33; 95% CI: 1.28–1.39, *p* < 0.001) and G3/G4 NETs (HR, 3.71, 95% CI: 3.59–3.83, *p* < 0.001) demonstrated worse OS. These findings were consistent in subgroup analyses of the distant GI NETs, pancreatic NETs, and pulmonary NETs.

After adjusting for other co-variates, regional disease (HR, 1.68, 95% CI: 1.63–1.73) and distant disease (HR, 4.46, 95%: 4.35–4.57, *p* < 0.001) showed worse OS than localized disease. According to the primary site, OS of liver NETs was the worst (HR, 1.75, 95% CI: 1.59–1.92), followed by the lung and pancreas (HR, 0.90, 95% CI: 0.87–0.93). The survival of appendiceal NETs was the best (HR: 0.55, 95% CI: 0.51–0.59). Based on race, black (HR: 1.15, 95% CI: 1.12–1.18, *p* < 0.001) and American Indian/Alaska Natives (HR: 1.20, 95% CI: 1.08–1.34, *p* = 0.001) were associated with worse OS compared to white.

## 4. Discussion

This updated population study using SEER-22 demonstrates that the incidence of NET nearly doubled from 4.6 per 100,000 persons in 2000 to 8.1 per 100,000 persons in 2021. A similar trend was observed in the landmark study by Dasari et al. [2], which reported an increase from 1.09 per 100,000 in 1973 to 6.98 per 100,000 in 2012 using SEER-18, as well as a recently published updated population study [5]. Stratification of the data revealed rising incidence rates across most primary sites, NET types, disease stages, and grades, with the most notable increases in G1/G2 and localized NETs. By primary site, pulmonary, colorectal, and small bowel NETs had the highest incidence rates, while pancreatic, appendiceal, and small bowel NETs showed the most significant increases from 2020 to 2021.

The overall increase in NET incidence can be attributed to multiple factors. Improved early detection, driven by advances in endoscopy, cross-sectional imaging, nuclear medicine diagnostic imaging, and possibly the availability of subspecialized pathologists over the past two decades, has significantly contributed to the rise in early-stage and low-grade NETs. Appendiceal NETs, in particular, have shown the most significant increase, especially from 2010 to 2015, a trend also observed in an epidemiology study in England from 2010 to 2013 with more than a two-fold increase [6]. This rise could be due to both increased appendectomy rates and the adoption of more extensive pathological assessments, such as routine histological examinations and thorough analysis of appendectomy specimens. Changes in the WHO classifications for neuroendocrine tumors in 2000/2004 and 2010 also played a role, with better understanding and characterization of NETs [6]. Additionally, the interplay between biological and environmental factors is significant, as evidenced by the higher incidence of gastric NETs in Asia, where it ranks as the second most common primary site in China and Japan, compared to the sixth highest in the present study (approximately 0.7 per 100,000 in 2021) [7,8].

The OS of NET patients has improved over time, with those diagnosed after 2015 demonstrating the best outcomes. Since the early 21st century, several landmark clinical trials have significantly impacted NET treatment and survival rates [3]. In 2011, the RADIANT-3 trial showed that everolimus improved progression-free survival (PFS) in patients with advanced pancreatic neuroendocrine tumors compared to placebo (median: 11 versus 4.6 months), while the SUN1111 trial found that sunitinib enhanced both PFS (median: 11.4 versus 5.5 months) and OS (median: 38.6 versus 29.1 months) in the same patient population [9,10]. The 2014 CLARINET trial demonstrated that lanreotide significantly prolonged PFS in patients with metastatic gastroenteropancreatic NETs (median was not reached at that time) [11]. This was followed by the 2016 RADIANT-4 trial, which showed that everolimus significantly improved PFS in patients with advanced, progressive, nonfunctional pulmonary and gastrointestinal NETs (median: 11.0 versus 3.9 months) [12]. The 2017 NETTER-1 trial confirmed that 177Lu-Dotatate provided superior PFS (median: not reached versus 8.4 months) and response rates compared to high-dose octreotide LAR in advanced midgut NET patients, with an 11.7-month difference in median OS reported in the final analysis in 2021 (median: 48.0 versus 36.3 months), though it was not statistically significant [13,14]. Later, the CAPTEM trial demonstrated that the combination of capecitabine and temozolomide was effective for advanced NET patients with manageable toxicity [15,16]. Nevertheless, it is prudent to say that several landmark, practice-changing studies over the past 10 years have incrementally improved outcomes of metastatic NET patients.

The combination of increased incidence, early detection, and improved prognosis has contributed to the rising prevalence of NETs during the study period [17,18]. Once considered an orphan disease, the latest incidence rate exceeding 8 per 100,000 persons suggests that NETs have become relatively common, comparable to multiple myeloma at 7.11 per 100,000 in 2020 and cervical cancer at 9.8 per 100,000 in 2019 [19,20]. With a more indolent disease course, functional subgroups, and increased prevalence, the demand for multidisciplinary NET programs is expected to rise. As patients live longer, attention and research should also focus on improving their quality of life [21,22,23,24]. Despite the overall increase in incidence and prevalence, NETs are a heterogeneous group of tumors originating from different primary sites, exhibiting varied behaviors and prognoses based on stage, grade, and differentiation, necessitating case-by-case management through a multidisciplinary approach [3,25,26,27]. Collaboration across medical specialties and institutions is crucial for both optimal clinical care and the collection of homogeneous samples for adequately powered clinical trials with high-level evidence.

The present study also captured the impact of COVID-19 on the incidence of NETs, noting a temporary decrease in 2020, especially among lower-grade and early-stage tumors, likely due to under-diagnosis. This trend mirrors observations in other cancer types, likely caused by limited resources for cancer screening and elective procedures during the pandemic [28,29,30]. By 2021, NET incidence rates rebounded, and it is anticipated that they will continue to rise in the post-COVID era as more data beyond 2022 becomes available.

The present study should be interpreted with several caveats. Firstly, this population database lacks clinically meaningful variables such as functional status, radiologic response, and treatment sequence. Secondly, the definition of NET has evolved over time, and diagnostic criteria from the early 2000s may not align with current guidelines. Variables like grading and differentiation were not carefully distinguished and recorded until the last five years. Additional inherent drawbacks of population databases include documentation errors, missing input, retrospective study design, and limited analysis of patient-level data. While the absolute incidence, prevalence, and survival rates may not accurately represent real-world data, the observed trends of increased incidence and improved prognosis hold more clinical significance.

## 5. Conclusions

The incidence of NETs has continued to rise in the United States, with current annual rates exceeding eight per 100,000—a threshold that challenges their historical classification as an orphan disease. Improved patient outcomes are likely driven by expanded access to effective diagnostic tools and therapeutic options. As a result, the growing prevalence of NETs is expected to increase the demand for comprehensive, multidisciplinary care programs and may catalyze further innovation in drug development.

While advanced-stage and high-grade NETs still require more effective treatment strategies, the majority of patients—those with indolent or functional NETs—may benefit most from a renewed clinical and research focus on long-term management, survivorship, and quality of care.

## Figures and Tables

**Figure 1 cancers-17-03323-f001:**
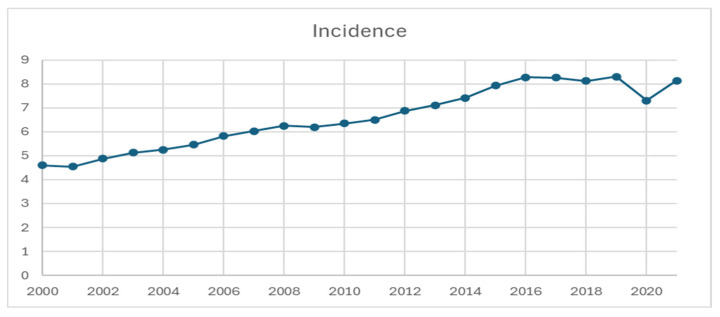
Incidence of neuroendocrine tumor from 2000 to 2021, per 100,000 persons.

**Figure 2 cancers-17-03323-f002:**
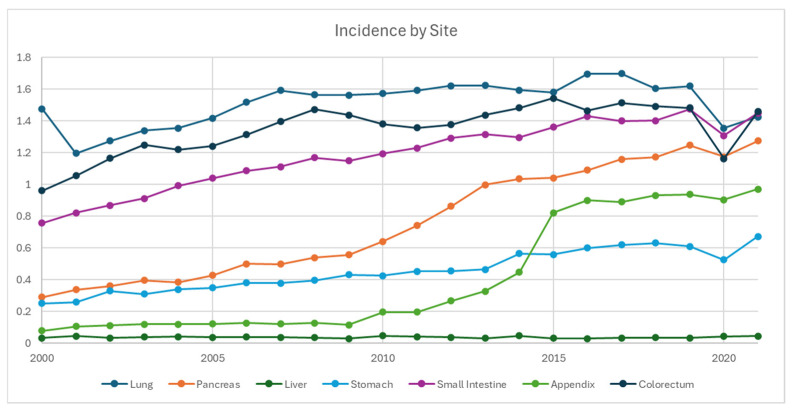
Incidence of neuroendocrine tumor from 2000 to 2021 stratified by primary site, per 100,000 persons.

**Figure 3 cancers-17-03323-f003:**
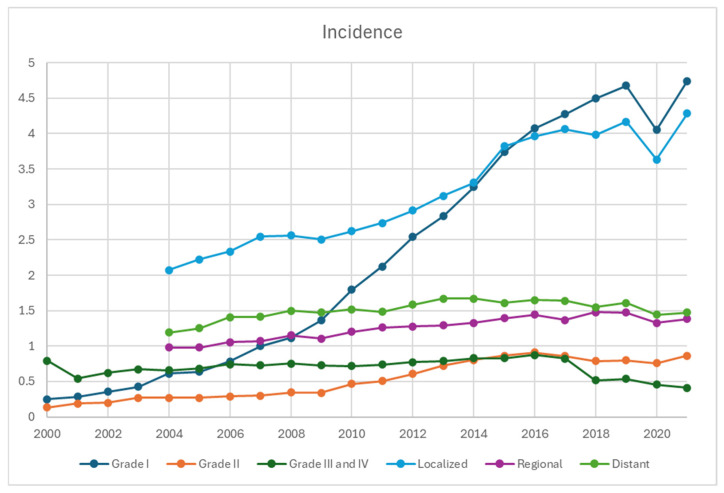
Incidence of neuroendocrine tumors from 2000 to 2021 stratified by grades and disease stage, per 100,000 persons. The earliest staging data available for SEER-22 database was 2004, accounting for the absence of datapoints in 2000–2003.

**Figure 4 cancers-17-03323-f004:**
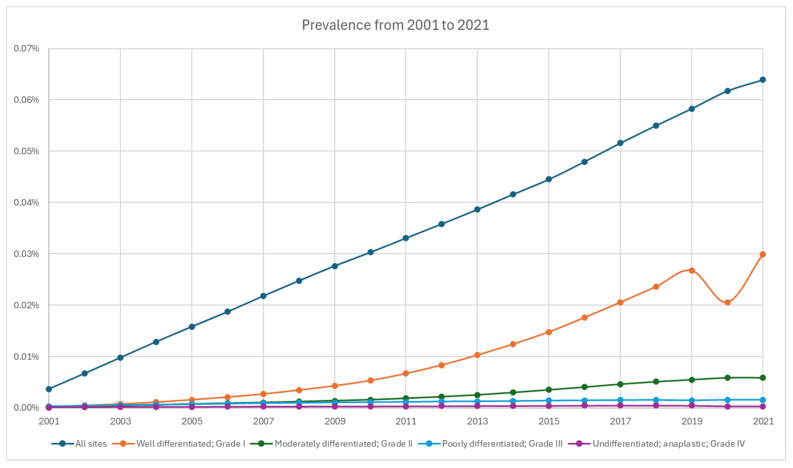
Prevalence of neuroendocrine tumor from 2001 to 2021 stratified by grade.

**Figure 5 cancers-17-03323-f005:**
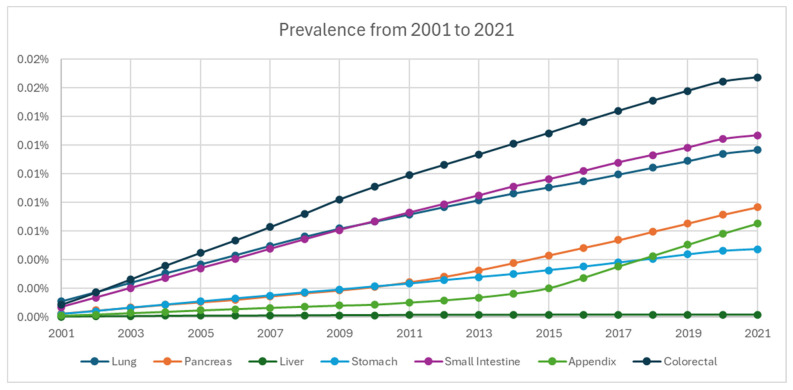
Prevalence of neuroendocrine tumors from 2001 to 2021 stratified by primary site.

**Figure 6 cancers-17-03323-f006:**
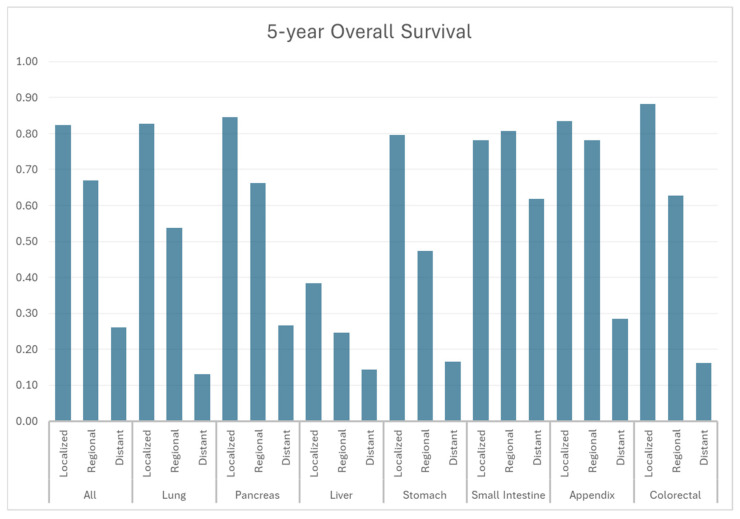
Five-year overall survival by stage and primary site.

**Figure 7 cancers-17-03323-f007:**
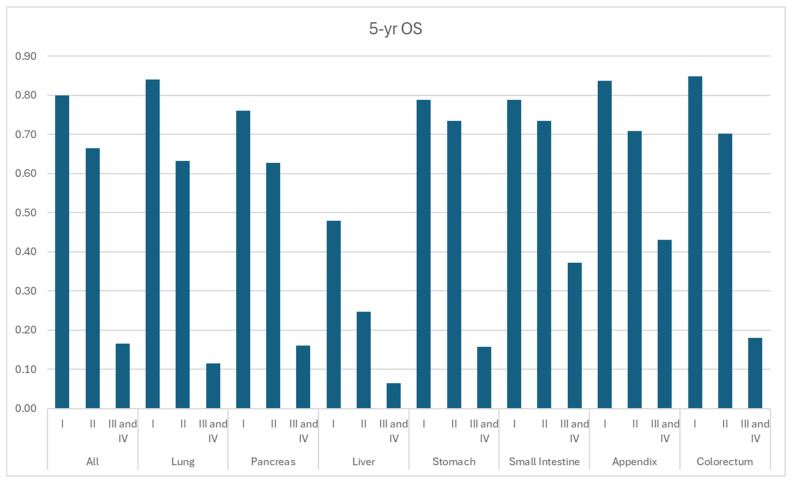
Year overall survival by grade and primary site.

**Table 1 cancers-17-03323-t001:** Incidence of neuroendocrine tumor from 2000 to 2021 stratified by age groups, per 100,000 persons.

d	Rate	Less Than 50 Years	50–64 Years	65 Years and Above
2000 Rate	4.599	1.183	9.767	18.029
2001 Rate	4.542	1.255	9.713	17.228
2002 Rate	4.876	1.26	10.875	18.462
2003 Rate	5.132	1.357	10.802	20.02
2004 Rate	5.252	1.335	11.402	20.378
2005 Rate	5.459	1.372	11.979	21.119
2006 Rate	5.816	1.471	12.743	22.476
2007 Rate	6.032	1.51	13.201	23.415
2008 Rate	6.247	1.591	13.765	23.985
2009 Rate	6.197	1.59	14.081	23.225
2010 Rate	6.352	1.676	13.87	24.204
2011 Rate	6.496	1.689	14.059	25.044
2012 Rate	6.879	1.826	14.987	26.189
2013 Rate	7.11	1.962	15.413	26.731
2014 Rate	7.409	2.112	16.027	27.51
2015 Rate	7.924	2.631	16.864	27.627
2016 Rate	8.279	2.69	17.183	29.716
2017 Rate	8.266	2.771	17.103	29.248
2018 Rate	8.117	2.763	16.901	28.348
2019 Rate	8.296	2.779	17.588	28.856
2020 Rate	7.307	2.594	14.912	25.265
2021 Rate	8.128	2.954	16.831	27.429

**Table 2 cancers-17-03323-t002:** Prevalence of neuroendocrine tumor based on SEER 22 database from 2001 to 2021.

Year	Prevalence	Count	Population	Duration (Year)
2001	0.00369%	4044	116,872,166	1
2002	0.00671%	7509	118,156,034	2
2003	0.00979%	11,191	119,330,102	3
2004	0.01287%	14,975	120,448,064	4
2005	0.01582%	18,748	121,440,491	5
2006	0.0187%	22,589	122,471,714	6
2007	0.02178%	26,811	123,703,952	7
2008	0.02475%	31,157	125,057,830	8
2009	0.02764%	35,589	126,480,118	9
2010	0.03029%	39,922	127,859,227	10
2011	0.03306%	44,536	129,189,098	11
2012	0.03582%	49,310	130,486,503	12
2013	0.03864%	54,341	131,716,649	13
2014	0.04159%	59,668	132,945,252	14
2015	0.04451%	65,270	134,221,529	15
2016	0.04796%	71,654	135,444,427	16
2017	0.05157%	78,382	136,518,227	17
2018	0.05499%	84,963	137,401,622	18
2019	0.05829%	91,468	138,133,638	19
2020	0.06174%	98,379	138,704,580	20
2021	0.06392%	103,221	138,877,544	21

**Table 3 cancers-17-03323-t003:** Survival by primary site and stage. NR: not reached.

Site	Stage	mOS (Months)	1-yr OS	3-yr OS	5-yr OS
All	All	NR	77.4%	66.1%	59.3%
	Localized	NR	94.4%	88.2%	82.3%
	Regional	NR	86.3%	74.7%	66.9%
	Distant	13.72	52.6%	34.0%	26.1%
Lung	All	56.0	67.7%	54.6%	49.1%
	Localized	NR	94.9%	88.5%	82.7%
	Regional	NR	79.3%	61.6%	53.8%
	Distant	6.87	35.1%	17.0%	13.0%
Pancreas	All	NR	79.6%	65.5%	56.4%
	Localized	NR	95.3%	89.9%	84.6%
	Regional	NR	87.2%	74.7%	66.2%
	Distant	20.96	60.9%	38.8%	26.7%
Liver	All	17.00	56.0%	36.1%	28.2%
	Localized	27.58	64.9%	48.6%	38.4%
	Regional	12.69	51.4%	34.0%	24.6%
	Distant	6.36	39.2%	21.9%	14.4%
Stomach	All	NR	85.9%	76.1%	69.0%
	Localized	NR	94.2%	86.4%	79.6%
	Regional	53.50	76.5%	58.0%	47.4%
	Distant	9.66	43.7%	22.2%	16.5%
Small Intestine	All	NR	91.4%	82.8%	74.4%
	Localized	NR	92.4%	85.2%	78.2%
	Regional	NR	94.0%	88.0%	80.8%
	Distant	NR	87.5%	73.9%	61.8%
Appendix	All	NR	91.0%	84.4%	75.9%
	Localized	NR	94.4%	90.5%	83.5%
	Regional	NR	93.4%	89.1%	78.1%
	Distant	29.47	72.8%	42.4%	28.5%
Colorectal	All	NR	84.8%	77.4%	72.1%
	Localized	NR	97.1%	93.1%	88.2%
	Regional	NR	83.8%	70.8%	62.7%
	Distant	8.49	41.6%	23.0%	16.2%

**Table 4 cancers-17-03323-t004:** Survival by primary site and grade. NR: not reached.

Site	Grade	mOS (Months)	1-yr OS	3-yr OS	5-yr OS
All	I	NR	93.8%	86.9%	80.0%
	II	NR	87.8%	75.8%	66.5%
	III and IV	9.51	43.3%	21.6%	16.6%
Lung	I	NR	95.6%	89.7%	84.1%
	II	NR	84.8%	71.2%	63.3%
	III and IV	8.29	38.7%	15.7%	11.5%
Pancreas	I	NR	92.8%	83.6%	76.0%
	II	NR	87.7%	72.6%	62.8%
	III and IV	8.88	41.3%	21.6%	16.0%
Liver	I	NR	85.6%	59.5%	48.0%
	II	14.16	54.7%	33.6%	24.6%
	III and IV	4.55	29.4%	11.7%	6.4%
Stomach	I	NR	94.0%	86.4%	78.9%
	II	NR	91.0%	81.3%	73.5%
	III and IV	9.84	44.2%	20.8%	15.8%
Small Intestine	I	NR	93.8%	86.7%	78.9%
	II	NR	92.6%	83.6%	73.4%
	III and IV	29.21	64.7%	47.3%	37.2%
Appendix	I	NR	94.6%	91.0%	83.8%
	II	NR	92.7%	82.3%	70.9%
	III and IV	46.74	76.3%	57.1%	43.1%
Colorectum	I	NR	95.0%	90.1%	84.8%
	II	NR	89.0%	78.7%	70.2%
	III and IV	8.87	41.1%	22.2%	18.0%

**Table 5 cancers-17-03323-t005:** Multivariable analysis of patients with neuroendocrine tumors diagnosed between 2000 and 2021. NA: not applicable.

	Overall (n = 120,827)		Distant GI NET (n = 8497)		Distant Pancreatic NET (n = 5533)		Distant Pulmonary NET (n = 10,657)	
	HR	*p*-Value	HR	*p*-Value	HR	*p*-Value	HR	*p*-Value
Year								
2000–2004	ref	ref	ref	ref	ref	ref	ref	ref
2005–2009	0.95 (0.93–0.98)	<0.001	1.00 (0.92–1.08)	0.996	0.97 (0.87–1.08)	0.591	0.98 (0.92–1.05)	0.660
2010–2014	0.97 (0.94–0.99)	0.017	1.07 (0.98–1.16)	0.123	0.90 (0.81–1.00)	0.050	0.96 (0.90–1.03)	0.292
2015–2021	0.92 (0.89–0.95)	<0.001	0.99 (0.90–1.08)	0.798	0.88 (0.80–0.98)	0.019	0.89 (0.93–0.95)	0.001
Grade								
I: Well differentiated	ref	ref	ref	ref	ref	ref	ref	ref
II: Moderately differentiated	1.33 (1.28–1.39)	<0.001	1.52 (1.37–1.69)	<0.001	1.20 (1.05–1.37)	0.009	2.23 (1.90–2.61)	<0.001
III and IV: Poorly differentiated and undifferentiated; anaplastic	3.71 (3.59–3.83)	<0.001	7.02 (6.45–7.63)	<0.001	3.19 (2.82–3.61)	<0.001	5.36 (4.73–6.06)	<0.001
Race								
White	ref	ref	ref	ref	ref	ref	ref	ref
American Indian/Alaska Native	1.20 (1.08–1.34)	0.001	1.09 (0.76–1.57)	0.649	1.10 (0.73–1.66)	0.643	1.02 (0.73–1.45)	0.874
Asian or Pacific Islander	0.94 (0.91–0.99)	0.010	1.23 (1.07–1.41)	0.003	1.07 (0.94–1.22)	0.298	0.96 (0.86–1.08)	0.503
Black	1.15 (1.12–1.18)	<0.001	1.14 (1.05–1.23)	0.002	1.15 (1.04–1.28)	0.007	1.08 (1.01–1.17)	0.029
Age								
≤30	ref	ref	ref	ref	ref	ref	ref	ref
31–60	2.30 (2.0–2.56)	<0.001	1.10 (0.81–1.49)	0.536	0.93 (0.74–1.18)	0.555	2.29 (1.52–3.46)	<0.001
≥61	5.38 (4.84–5.98)	<0.001	2.00 (1.48–2.70)	<0.001	1.48 (1.17–1.87)	0.001	3.04 (2.02–4.59)	<0.001
Stage								
Localized	ref	ref	NA	NA	NA	NA	NA	NA
Regional	1.68 (1.63–1.73)	<0.001	NA	NA	NA	NA	NA	NA
Distant	4.46 (4.35–4.57)	<0.001	NA	NA	NA	NA	NA	NA
Site								
Lung	ref	ref	NA	NA	NA	NA	NA	NA
Appendix	0.55 (0.51–0.59)	<0.001	NA	NA	NA	NA	NA	NA
Colorectal	0.54 (0.52–0.56)	<0.001	NA	NA	NA	NA	NA	NA
Liver	1.75 (1.59–1.92)	<0.001	NA	NA	NA	NA	NA	NA
Pancreas	0.90 (0.87–0.93)	<0.001	NA	NA	NA	NA	NA	NA
Small bowel	0.61 (0.59–0.63)	<0.001	NA	NA	NA	NA	NA	NA
Stomach	0.83 (0.80–0.86)	<0.001	NA	NA	NA	NA	NA	NA

## Data Availability

The data underlying this article are available in SEER.

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
