# Peer review of "Dethroning of Neuroendocrine Tumor as an Orphan Disease: US Incidence, Prevalence, and Survival in the 21st Century"

_cancers, 2025, doi:10.3390/cancers17203323_

Round 1

Reviewer 1 Report

Comments and Suggestions for Authors

A well written overview from the  SEER data. Nothing new or original as  much of this has been presented aand seen before. However as a review for the  general reader it reads well and  highlights the changes  in incidence/prevalence.

The new updated paper from Dasari could be added. JAMA 2025

https://jama.jamanetwork.com/article.aspx?doi=10.1001/jamanetworkopen.2025.15798&utm_campaign=articlePDF%26utm_medium=articlePDFlink%26utm_source=articlePDF%26utm_content=jamanetworkopen.2025.15798

Page 2 introduction it is Cu64 not 65 as in the brackets.  Incidentally why picj Cu64 when there are Al compounds and  agonists

The whole question of  goblet cells vs MANECs/MINENs is really obsolete as  most authorities reject these as NENs

Hepatic NENs are really rare and  just reflect old  imaging failing to detect the likely primary.

The rising incidence is clearly genuine  but many other factors including specialist pathologists. I suspect many were wrongly labelled in past

In the discussion I think the  impact of RADIOANT and SUN111 studies  has been overemphasised. The PFS were only 11 months whereas NETTER 1 substantially  better

The authors fall into trap of NENs vs NETs vs NECs. Much of the paper refers to NETs where it should be NENs, explain the difference please. 

See 2nd para in conclusion "high grade NETS" i suspect they mean G3NEC rather than G3NET

This and the Cu64/65 indicate some sloppiness

Overall with some  minor rewriting it is  publishable

Author Response

We would like to extend our gratitude and appreciation to all of you for taking the time to read and review our work so thoroughly. We greatly value your comments and suggestions and we are thankful for the opportunity to improve our manuscript. Please find below the responses and changes made based on your comments and queries.

1.The new updated paper from Dasari could be added. JAMA 2025

https://jama.jamanetwork.com/article.aspx?doi=10.1001/jamanetworkopen.2025.15798&utm_campai gn=articlePDF%26utm_medium=articlePDFlink%26utm_source=articlePDF%26utm_content=jaman etworkopen.2025.15798

-This cited study was published two months ago after the present manuscript was written. We added this reference as a citation, which showed similar trends: “…This updated population study using SEER-22 demonstrates that the incidence of NET nearly doubled from 4.6 per 100,000 persons in 2000 to 8.1 per 100,000 persons in 2021. A similar trend was observed in the landmark study by Dasari et al.2, which re-ported an increase from 1.09 per 100,000 in 1973 to 6.98 per 100,000 in 2012 using SEER-18, as well as a recently published updated population study5”

2.Page 2 introduction it is Cu64 not 65 as in the brackets.  Incidentally why picj Cu64 when there are Al compounds and  agonists

-Revised accordingly.

3.The whole question of  goblet cells vs MANECs/MINENs is really obsolete as  most authorities reject these as NENs

-deleted this statement per reviewer recommendations.

4.Hepatic NENs are really rare and  just reflect old  imaging failing to detect the likely primary.

-Agree with reviewer. We characterized the liver NETs in order to be consistent with the original paper by James Yao who reported its outcomes.

Yao JC, Hassan M, Phan A, et al. One hundred years after “carcinoid”: epidemiology of and prognostic factors for neuroendocrine tumors in 35,825 cases in the United States. Journal of clinical oncology. 2008;26(18):3063-3072.

5.The rising incidence is clearly genuine  but many other factors including specialist pathologists. I suspect many were wrongly labelled in past

-Agree. We have made the following revisions to take this factor into consideration:

“The overall increase in NET incidence can be attributed to multiple factors. Improved early detection, driven by advances in endoscopy, cross-sectional imaging, nuclear medicine diagnostic imaging, and possibly availability of subspecialized pathologists over the past two decades, has significantly contributed to the rise in early-stage and low-grade NETs.”

6.In the discussion I think the  impact of RADIOANT and SUN111 studies  has been overemphasised. The PFS were only 11 months whereas NETTER 1 substantially  better

-In order to avoid such overemphasis, we removed verbiage such as “significantly” and added numeric values for PSF and OS:

“The OS of NET patients has improved over time, with those diagnosed after 2015 demonstrating the best outcomes. Since the early 21st century, several landmark clini-cal trials have significantly impacted NET treatment and survival rates3. In 2011, the RADIANT-3 trial showed that everolimus significantly improved progression-free sur-vival (PFS) in patients with advanced pancreatic neuroendocrine tumors compared to placebo (median: 11 versus 4.6 months), while the SUN1111 trial found that sunitinib significantly enhanced both PFS (median: 11.4 versus 5.5 months) and OS (median: 38.6 versus 29.1months) in the same patient population9,10. The 2014 CLARINET trial demonstrated that lanreotide significantly prolonged PFS in patients with metastatic gastroenteropancreatic NETs (median was not reached at that time)11. This was fol-lowed by the 2016 RADIANT-4 trial, which showed that everolimus significantly im-proved PFS in patients with advanced, progressive, nonfunctional pulmonary and gas-trointestinal NETs (median: 11.0 versus 3.9 months) 12. The 2017 NETTER-1 trial con-firmed that 177Lu-Dotatate provided superior PFS (median: not reached versus 8.4 months) and response rates compared to high-dose octreotide LAR in advanced midgut NET patients, with an 11.7-month difference in median OS reported in the final analy-sis in 2021 (median: 48.0 versus 36.3 months), though it was not statistically significant13,14. Later, the CAPTEM trial demonstrated that the combination of capecita-bine and temozolomide was effective for advanced NET patients with manageable tox-icity15,16. Nevertheless, it is prudent to say that several landmark practice changing studies over past 10 years have incrementally improved outcomes of metastatic NET patients.”

7.The authors fall into trap of NENs vs NETs vs NECs. Much of the paper refers to NETs where it should be NENs, explain the difference please. 

-for the purpose of this study, we adopted the general term of “NET” to be consistent with the study methodology, selection criteria, and nomenclature of the two prior studies:

1.Dasari A, Shen C, Halperin D, Zhao B, Zhou S, Xu Y, Shih T, Yao JC. Trends in the incidence, prevalence, and survival outcomes in patients with neuroendocrine tumors in the United States. JAMA oncology. 2017 Oct 1;3(10):1335-42.

2.Yao JC, Hassan M, Phan A, Dagohoy C, Leary C, Mares JE, Abdalla EK, Fleming JB, Vauthey JN, Rashid A, Evans DB. One hundred years after “carcinoid”: epidemiology of and prognostic factors for neuroendocrine tumors in 35,825 cases in the United States. Journal of clinical oncology. 2008 Jun 20;26(18):3063-72.

8.See 2nd para in conclusion "high grade NETS" i suspect they mean G3NEC rather than G3NET

-Yes. Either terminology can be used.

9.This and the Cu64/65 indicate some sloppiness

 -Revised

Reviewer 2 Report

Comments and Suggestions for Authors

This paper is dealing with the incidence, prevalence and survival of neuroendocrine tumors (NETs) in the US from 2000 to 2021. Data of 231,659 patients were included suffering from NETs of various organs. During this time the incidence doubled, especially grade 1 tumors increased and in addition also the survival times. Thus, a main conclusion of this paper is that NETs are no longer an orphan disease.

The methods and statistical data are clearly presented in figures and tables. The according text is well understandable for the reader. The data are interpreted sufficiently and with caution.

Criticism:

 Page 7: The numbers of figures 5 and 6, and also tables 3A and 3 B are mixed up.

 Page 10:  The table is named 17 but referred to in the text as Table 3B.

Page 3:  The year 2010 is given but most probably 2000 is correct.  

These points should be clarified.

Author Response

We would like to extend our gratitude and appreciation to all of you for taking the time to read and review our work so thoroughly. We greatly value your comments and suggestions and we are thankful for the opportunity to improve our manuscript. Please find below the responses and changes made based on your comments and queries.

1.Page 7: The numbers of figures 5 and 6, and also tables 3A and 3 B are mixed up.

-Revised and re-labeled accordingly.

2. Page 10:  The table is named 17 but referred to in the text as Table 3B.

-Revised accordingly. It should read “SEER 17”.

3.Page 3:  The year 2010 is given but most probably 2000 is correct.  

 -Agree. Revised accordingly.